# Genomic Profiling of Multidrug-Resistant Swine *Escherichia coli* and Clonal Relationship to Human Isolates in Peru

**DOI:** 10.3390/antibiotics12121748

**Published:** 2023-12-18

**Authors:** Luis Alvarez, Dennis Carhuaricra, Joel Palomino-Farfan, Sonia Calle, Lenin Maturrano, Juan Siuce

**Affiliations:** 1Laboratory of Veterinary Bacteriology and Mycology, Faculty of Veterinary Medicine, National University of San Marcos, Lima 15021, Peru; luis.alvarez2@unmsm.edu.pe (L.A.); jpalominof@unmsm.edu.pe (J.P.-F.); scallee@unmsm.edu.pe (S.C.); 2Research Group in Biotechnology Applied to Animal Health, Production and Conservation [SANIGEN], Laboratory of Biology and Molecular Genetics, Faculty of Veterinary Medicine, National University of San Marcos, Lima 15021, Peru; heraud04@gmail.com (D.C.); amaturranoh@unmsm.edu.pe (L.M.)

**Keywords:** *Escherichia coli*, MDR, swine, human, phylogenetics

## Abstract

The misuse of antibiotics is accelerating antimicrobial resistance (AMR) in *Escherichia coli* isolated from farm animals. The genomes of ten multidrug-resistant (MDR) *E. coli* isolates from pigs were analyzed to determine their sequence types, serotypes, virulence, and AMR genes (ARGs). Additionally, the relationship was evaluated adding all the available genomes of Peruvian *E. coli* from humans using the cgMLST + HierCC scheme. Two aEPEC O186:H11-ST29 were identified, of which H11 and ST29 are reported in aEPEC isolates from different sources. An isolate ETEC-O149:H10-ST100 was identified, considered a high-risk clone that is frequently reported in different countries as a cause of diarrhea in piglets. One ExPEC O101:H11-ST167 was identified, for which ST167 is an international high-risk clone related to urinary infections in humans. We identified many ARGs, including extended-spectrum β-lactamase genes, and one ETEC harboring the *mcr-1* gene. CgMLST + HierCC analysis differentiated three clusters, and in two, the human isolates were grouped with those of swine in the same cluster. We observed that Peruvian swine MDR *E. coli* cluster with Peruvian *E. coli* isolates from healthy humans and from clinical cases, which is of great public health concern and evidence that AMR surveillance should be strengthened based on the One Health approach.

## 1. Introduction

Post-weaning disease (PWD) caused by *Escherichia coli* produces weight loss and death in recently weaned piglets, causing economic losses worldwide [1,2], and the misuse of antibiotics in animal farms is increasing multidrug resistance (MDR) to enterobacteria such as *E. coli* [3]. In the last decade, an increase in the incidence of outbreaks of severe sudden diarrhea associated with *E. coli* in piglets has been observed worldwide; with isolates frequently resistant to three or more antibiotic classes [4]. In particular, the emergence of CTX-M-type extended-spectrum β-lactamase (ESBL) and carbapenemase-producing *E. coli* strains has raised significant concern, warranting attention in surveillance studies of antimicrobial resistance (AMR) [5,6]. Furthermore, it has been shown that infections in humans may be related to strains isolated from pigs, suggesting that these originated at pig farms, or due to contamination during the production process [7].

Assessing the diversity and distribution of resistance genes in bacterial populations is a very useful tool to better understand the epidemiology of AMR [8]. It has been reported that northern hemisphere animal farm *E. coli* isolates most often show resistance to tetracyclines, sulfonamides, and streptomycin or spectinomycin [9,10]. In Peru, no current epidemiological surveillance plan for AMR exists that includes bacterial populations isolated from farm animals, so analysis or evaluation of their relationship with human isolates is not possible [11].

In this study, we characterized a collection of ten Peruvian *E. coli* MDR strains using genomic analysis to determinate their pathogenicity, serotype, sequence type, and the occurrence of AMR. We also analyzed the relationship between the porcine and human *E. coli* isolates from Peru.

## 2. Results

### 2.1. Serotype and Muli-Locus Sequence Typing (MLST)

The O serogroup of eight isolates was identified from genome sequence data, while the two remaining isolates were unassignable (O serogroup untypable). The H serogroup of 10 isolates was also sequenced. Overall, seven different O types and nine H types were identified, and the assigned serogroups for each isolate are detailed in Appendix A. Using the Achtman seven-locus MLST scheme, we identified six different sequence types (ST) from genome sequence data: ST-10 (n = 4), ST-29 (n = 2), ST-100 (n = 1), ST-167 (n = 1), ST-1602 (n = 1), and ST-3345 (n = 1) (Appendix A).

### 2.2. Virulence and Resistance Genes

Two *E. coli* isolates were *eae*-positive and *bfp*-negative, indicating that they belong to the atypical enteropathogenic *E. coli* (aEPEC) pathotype. One *E. coli* isolate had *lt*, *stb*, and *F4* genes, associated with enterotoxigenic *E. coli* (ETEC). One isolate had a different set of virulence genes including *pap* (Adhesins), *fyuA*, *iucC*, *irp2*, *iutA*, *sitA* (Iron uptake), *ompA*, *iss*, and *traT* (Protectins/serum resistance) associated with extraintestinal pathogenic *E. coli* (ExPEC) (Appendix A).

Few virulence genes were detected in the remaining isolates, and they were not classified as pathogenic. The aEPEC isolates presented around 90 virulence genes related to the locus for enterocyte effacement (LEE) translocator (*espA*, *espB*, *espD*, *tir*), type III secretion system (esc), and non-LEE effector (*nleA*, *nleB*, *nleE*) alleles (Appendix A).

We identified 31 antibiotic resistance genes (ARGs) (Figure 1a) belonging to 10 different antibiotic classes. The amount of ARG per isolate is detailed in Table 1. These genes confer resistance to 10 different classes of antibiotics (aminoglycosides, tetracyclines, macrolides, amphenicols, quinolones, sulfonamides, beta-lactams, trimethoprim, lincosamides, polymyxins). Eight isolates carried an ESBL, including *bla*_TEM-1B_ (n = 4), *bla*_TEM-1A_ (n = 2), *bla*_CTX-M-14_ (n = 1), and *bla*_TEM-176_ (n = 1). The results of the antibiotic sensitivity test are available in Appendix A.

In addition, one isolate (EC204K) carried the *mcr-1* gene that confers resistance to colistin (see Figure 1a and Appendix A). The *mcr-1* sequence of the *E. coli* EC204K isolate was 100% identical to the *mcr-1* sequence from the Resfinder database. The analysis of the genetic context of *mcr-1* allowed us to identify *nikB–mcr1–pap2* structure within IncI2 plasmid.

### 2.3. Population of Swine and Human Peruvian E. coli Isolates

As detailed in the methods section, all human *E. coli* genomes available from the Peruvian EnteroBase were included in the analysis. We used the EnteroBase *Escherichia* cgMLST + HierCC (2513 genes) scheme, which assigns bacterial genomes to hierarchical clusters (HCs). According to cgMLST + HierCC, 51% of the isolates (n = 154) were grouped in CgST Cplx 13, equivalent to Clonal Complex 10 (CC10) of the MLST scheme. The remaining isolates (n = 148) belong to 54 different CgST Cplx. Additionally, it was possible to differentiate three clusters (Figure 2a). Cluster 1 (n = 164), surrounded by a purple line, contains mostly isolated CgST Cplx 13. Cluster 2 (n = 80), surrounded by a yellow line, contains 23 different CgST Cplx. Cluster 3 (n = 58), surrounded by a red line, contains 22 different CgST Cplx.

Regarding the origin of the isolates, 80% of the porcine isolates belonged to CgST Cplx 13 (CC10) and grouped in cluster 1 with the human isolates. The remaining 20% of the porcine isolates grouped with other human isolates in cluster 2. Cluster 3 only contained human isolates (Figure 2b).

Concerning the pathotypes, 25 isolates of the ETEC pathotype were distributed in clusters 1 and 2 (see Appendix A). In cluster 1, one isolate was of porcine origin and eighteen were of human origin, while in cluster 2, all six isolates were of human origin. For cluster 1, the closeness of human ETEC isolates and swine ETEC was observed, so a maximum likelihood tree was constructed based on a single-nucleotide polymorphism (SNP) matrix (Figure 3a). The SNP matrix consisted of 7473 SNP positions (see Appendix A). The minimum SNP difference between the swine and human ETEC isolates was 3472. 

In regard to EPEC, 32 isolates were distributed in all clusters. In cluster 1, sixteen isolates were of human origin, while in cluster 2, two isolates were of porcine and fourteen of human origin, and in cluster 3, all five isolates were of human origin. For cluster 2, given the closeness of the human EPEC and EHEC to the swine EPEC isolates, a maximum likelihood tree based on a SNP matrix for 3991 positions was constructed (Appendix A) (Figure 3b). The minimum SNP difference between one of the EPEC swine and human EHEC isolates was 2455.

Finally, regarding the phylogroups according to Clermont type (see Appendix A), almost all the isolates of phylogroup A belonged to cluster 1, with only two isolates of the phylogroup in cluster 3. All the isolates of phylogroup B1 and C were grouped in cluster 2, while the isolates of phylogroups B2, D, E, F, and G were grouped in cluster 3. In addition, one isolate from cluster 1 was classified as unknown.

## 3. Discussion

The relationship between Enterobacteriaceae isolates from farm animals and isolates from humans has been reported in several countries, showing that animals can be a source of potentially dangerous bacteria for humans [3]. The objective of this study, therefore, was to determine genomic profiling of MDR swine *E. coli* and the clonal relationship to human isolates in Peru, analyzing the complete genome of 10 MDR *E. coli* isolated from pigs.

During characterization of the isolates in this study, we identified one belonging to the ETEC-O149:H10-ST100 clonal lineage that has frequently been isolated and identified as the cause of PWD in pigs [12,13]. Since it was first reported in 2013, the O149:H10-ST100 strain has been considered a high-risk clone due to its rapid geographic dispersion and dominance, high antibiotic resistance, and pathogenicity [14]. In addition, two isolates were identified as aEPEC O186:H11-ST29. Both ST29 and serogroup H11 have been reported in aEPEC isolates from humans, meat, and animals [15,16,17]. One isolate was identified as ExPEC O101:H11-ST167, with ST167 an international high-risk clone associated with cases of urinary tract infections in humans and isolated from fecal samples of hospital patients [18,19,20]. ST167 has been reported from pig samples in India and Germany [21,22,23], but this is the first report of this clone from pig samples in Peru. The characterization of these bacteria allows us to note that pathogenic *E. coli* isolates from farm animals, such as pigs, may share a serotype, serogroup, or sequence type with *E. coli* isolates obtained from human clinical samples. Furthermore, these results suggest that, as in studies from countries in Europe and the United States, domestic animals are natural reservoirs of pathogenic strains of *E. coli* that can be transmitted to humans through direct contact or through food contamination [24,25]. It is even more worrisome when the strains isolated from animals belong to internationally high-risk clones such as ST167, as recently reported in a 66-year-old man in the United States who had pyelonephritis caused by a *E. coli* ST167 *Bla*_NDM-5_ with resistance to all available β-lactams [26]. Unfortunately, we have sequenced too few isolates to really know the extent of dissemination of these high-risk clones in Peru. Therefore, it is of utmost importance to analyze the porcine *E. coli* population more widely and document the current situation of these high-risk clones.

Regarding AMR, we identified 96 ARGs in the 10 isolates, including ESBL genes, and the high-risk clone ETEC-O149:H10-ST100 harboring the *mcr-1* gene. The *mcr-1* gene was located between the *nikB* and *pap2* genes and carried an IncI2 plasmid (Figure 1b). The *nikB-mcr-1-pap2* structure is conserved among IncI2 plasmids, as previously reported in isolates from poultry in Peru [27] and other countries in Latin America [28]. The prevalence of positive isolates for the *mcr-1* gene from pig samples has been reported in China (20.6%), Vietnam (22%), and Belgium (13.2%) [29,30,31]. In Peru, a recent study found 12.5% (18/144) of isolates from pig farms positive for the *mcr-1* gene [27]. 

The global emergence of *mcr-1* isolates [32,33,34,35] can be explained by its unregulated use as a therapeutic drug and growth promoter in pig farms, leading to the appearance of colistin-resistant strains [36]. In December 2019, the use of colistin in farm animals was banned in Peru [37], and it is expected that this ban will have an effect on the levels of resistance to this antibiotic.

All the isolates in this study carried at least one gene for four different classes of antibiotics (tetracyclines, florfenicol, sulfonamides, and macrolides). This is consistent with studies at pig farms in China, Thailand, and Spain, where 90.5% (1694/1871), 91.3% (137/150), and 87% (163/187) of MDR *E. coli* were reported, respectively [38,39,40]. High levels of AMR would be explained by the inappropriate use of antibiotics in animal production, which generates selective pressure on the bacteria that colonize production animals, thus generating an increase in antibiotic resistance reports [41,42]. In addition, risk factors such as the introduction of new animals on farms and poor hygiene measures are associated with the emergence of MDR bacteria [43]. However, the information about the use of antibiotics on farms available from low- and middle-income countries (LMICs), such as Peru, is insufficient to permit the correlation between AMR and antibiotic misuse [44].

To determine the clonal relationships of the porcine *E. coli* isolates in the present study, we included all human isolates available in the Peruvian EnteroBase. These included results from two studies, one analyzed fecal samples from healthy people who sold chicken meat and others who did not, as well as babies from the main market and surrounding areas of the Villa El Salvador district in Lima [45], while the second evaluated changes in the intestinal microbiome of visitors living in Cusco for at least two months [46]. We included the analysis of *E. coli* isolates from healthy humans in the first study, and from the second study, the isolates from visiting humans with diarrhea acquired during their stay in Cusco.

Analysis using the cgMLST + HierCC scheme differentiated three clusters. Cluster 1 is the largest, composed mainly of cgST Cplx13 isolates (Equivalent to CC10) belonging to phylogroup A and containing isolates of human and swine origin, nineteen and sixteen of which belong the ETEC and EPEC type, respectively. As previously reported, phylogroup A and CC10 are dominant in the intestinal population of animals and humans [47,48]. Our results are also consistent with those of Reid et al. [49], who analyzed 248 genomes of CC10 *E. coli* isolates available in the EnteroBase. They found that CC10 has a worldwide distribution, is obtained from different sources, and can harbor intestinal and extra-intestinal pathogenic *E. coli* with a high level of AMR [49].

Cluster 2, the second largest, is composed of eleven different cgST Cplx isolates, belonging mostly to phylogroup B1, and containing 22 pathogenic isolates: 2 swine EPEC isolates, 14 human EPEC isolates, and 6 human ETEC isolates. Regarding the distribution of pathotypes, our analysis agrees with the studies by Sahl et al. [50] and Hao et al. [51] who showed that isolates of the same pathotype can belong to different phylogroups. In cluster 1, we observed a swine ETEC with a 3472 SNP difference from human ETEC, reflecting a moderate relationship. In cluster 2, a similar case with a SNP distance of 2455 between swine and human EPEC may also indicate a moderate relationship. ETEC, EPEC, and EHEC were isolated from the group of visitors to Cusco, so we hypothesize that the ETEC and EPEC isolates from pigs located in Lima could have been imported to Cusco some time ago, since almost half of Peruvian pig production is located there [27]. However, more data on the Peruvian population of *E. coli* in pigs and humans are needed to obtain a better overview.

Cluster 3 is the most diverse, comprising eleven different cgST Cplx isolates, belonging to phylogroups A, B2, D, E, F, and G from human isolates. The only isolate of phylogroup A, analyzed under the cgMLST scheme, belongs to a different cgST Cplx than the isolates of cluster 1, showing a higher level of discrimination of this scheme [52,53]. Therefore, the swine *E. coli* isolates analyzed in this study were not grouped into phylogroups other than A and B1 [40].

In conclusion, we observed that Peruvian isolates of porcine *E. coli* had diverse virulence and resistance gene profiles, and they grouped with Peruvian isolates of *E. coli* from both healthy humans and clinical cases, and included international high-risk clones. It is important to highlight that the high level of resistance to various classes of antibiotics found in strains of porcine *E. coli* is of great concern, being reservoirs of resistance genes that can be spread throughout the food chain, threatening public health. The *E. coli* pathotype sequences will be important for epidemiological studies of *E. coli* strains from Peruvian swine. Finally, it is suggested that surveillance systems for AMR in LMICs must be strengthened and adapted to the One Health approach, not limited to human health reports from hospitals, but also including farms, the production chain of food, and the environment.

## 4. Materials and Methods

### 4.1. Isolation, Sequencing, and Genome Sequences

The data of 10 Peruvian porcine *E. coli* strains included in this analysis are available as raw reads in SRA and GenBank [54]. The details of the isolation and sequencing methodologies have previously been published [54], but are briefly detailed here. 

During 2020, intestinal tissue samples were taken from diseased piglets (approximately 8 weeks old) for the isolation of *E. coli* strains. Samples were inoculated on MacConkey selective agar and incubated for 24 h at 37 °C. Lactose-positive colonies were analyzed by conventional biochemical testing for confirmation of *E. coli* [55]. Antimicrobial susceptibility testing was carried out using the Kirby–Bauer disk diffusion method following Clinical and Laboratory Standards Institute guidelines [56]. The 10 isolates with the highest resistance to antibiotics were selected for sequencing.

DNA was extracted using the GeneJET genomic DNA purification kit (Thermo Fisher Scientific, Waltham, MA, USA) following the manufacturer’s instructions. DNA libraries were prepared following the Nextera XT DNA library preparation kit instructions (Illumina, San Diego, CA, USA) and loaded on an illumina MiSeq instrument using the 2 × 250-bp paired-end format. Post-sequencing, filtering of low-quality raw reads, and de novo assembly were performed using Trimmomatic v0.36 [57] and the SPAdes v3.13.0 genome assembler [58], respectively.

For the cgMLST analysis, 292 genomes of Peruvian *E. coli* isolates from humans available in EnteroBase were also included (https://enterobase.warwick.ac.uk/species/index/ecoli, accessed on 30 April 2023).

### 4.2. Genome Characterization

We used the tools MLST v.2.0 [59] and SerotypeFinder v.2.0 [60] from the Center for Genomic Epidemiology (CGE) (http://www.genomicepidemiology.org, accessed on 15 July 2023) to determine the sequence types and serotypes of the isolates. The ClermonTyping tool (http://clermontyping.iame-research.center/, accessed on 10 August 2023) was used to predict the phylogroups [61,62]. CGE and ClermonTyping tool predictions were called using default settings. ARGs and virulence genes were annotated using the Resfinder and VirulenceFinder databases from CGE with the ABRICATE v.1.0.1 tool (https://github.com/tseemann/abricate, accessed on 22 July 2023) at 80% nucleotide identity and 80% minimum coverage as parameters. The *mcr-1* gene context was examined in the Artemis [63] genome browser and drawn with Easyfig [64]. The plasmid replicon type was predicted using the Plasmidfinder v2.1 [65] webserver (https://cge.food.dtu.dk/services/PlasmidFinder/, accessed on 12 November 2023).

### 4.3. CgMLST Hierarchical Clustering Analysis

Raw sequence data files of the 10 isolates were uploaded to EnteroBase for cgMLST analysis and to produce a cgST (https://enterobase.warwick.ac.uk/species/index/ecoli, accessed on 30 April 2023). In addition, the EnteroBase database was queried to select all available Peruvian *E. coli* recovered from humans. These were included in the cgMLST + HierCC analysis with the 10 porcine isolates from the present study. Briefly, cgMLST V2 is + HierCC defines clusters based on cgMLST. Distance between genomes is calculated using the number of shared cgMLST alleles. For *E. coli*, it conglomerates into 11 levels, and the HC1100 level is where the ST complexes (cg Cplx) are recognized, and is the level being used for analysis in this study [52,66]. That is, if two isolates differ from each other by less than 1100 alleles, they both belong to the same hierCC (HC1100). A tree was inferred using GrapeTree of EnteroBase based on the Neighbor-joining (RapidNJ) algorithm [67].

CSI Phylogeny v.1.4 [68] from the CGE (http://www.genomicepidemiology.org, accessed on 25 July 2023) was used to identify single SNPs between swine and human origin isolates that were close to each other. The *E. coli* Strain K-12 substr. MG1655 genome was included as a reference strain, and CGE default parameters were used during SNP analysis. The phylogenetic tree was visualized and edited by using the bioinformatics tool iTOL v6 [69].

## Figures and Tables

**Figure 1 antibiotics-12-01748-f001:**
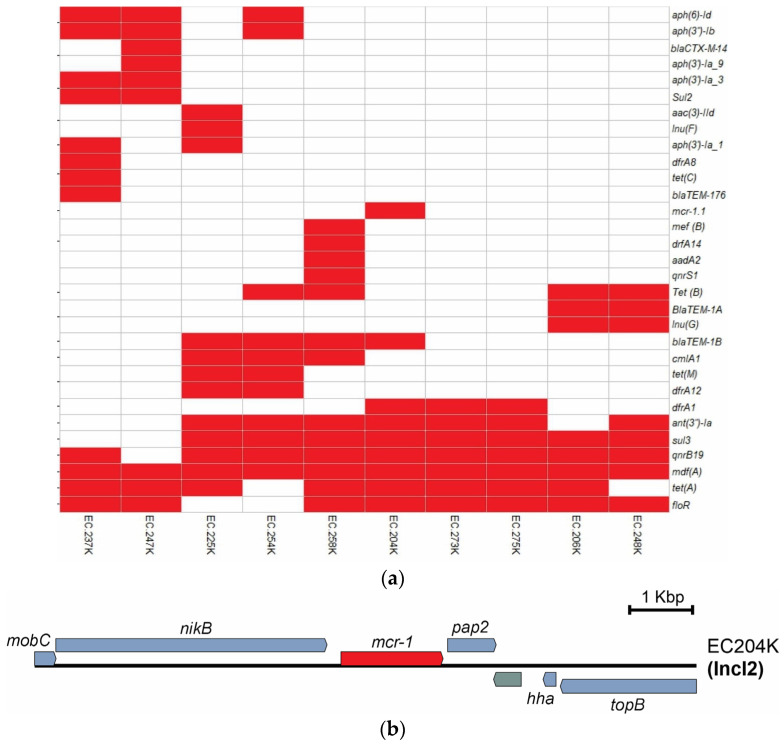
Analysis of ARGs from swine *E. coli* isolates. (**a**) Heat map (presence/absence) of ARGs by class of antibiotics. (**b**) Genetic context of the *mcr-1* gene of swine *E. coli* isolate EC204K. The *mcr-1* gene is marked in red. *pap2*, *mobC*, *topB*, *hha*, and *nickB* genes are marked in blue.

**Figure 2 antibiotics-12-01748-f002:**
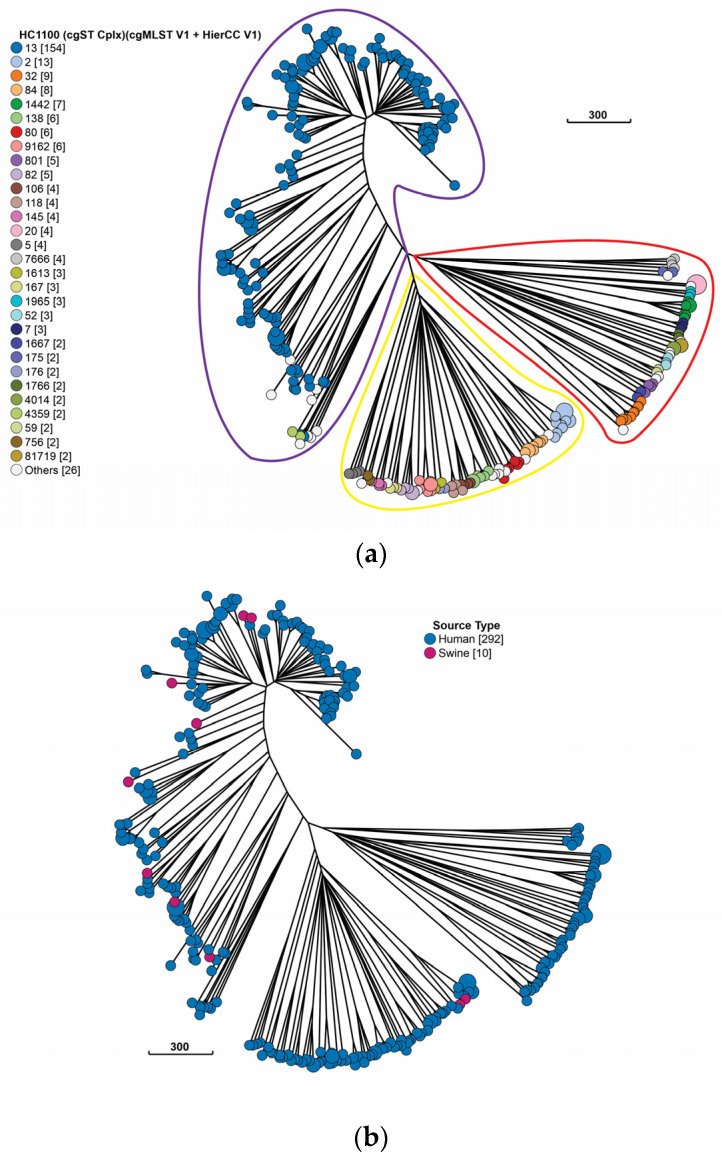
GrapeTree inferred using the Neighbor-joining (RapidNJ) algorithm based on the cgMLST V1 + Hierarchical Clustering (HierCC) V1 scheme of 302 Peruvian genomes according to the following: (**a**) HC1100(CgST Cplx); (**b**) according to isolation source. The numbers in brackets are the number of isolates belonging to a cgST Cplx (**a**) or source type (**b**).

**Figure 3 antibiotics-12-01748-f003:**
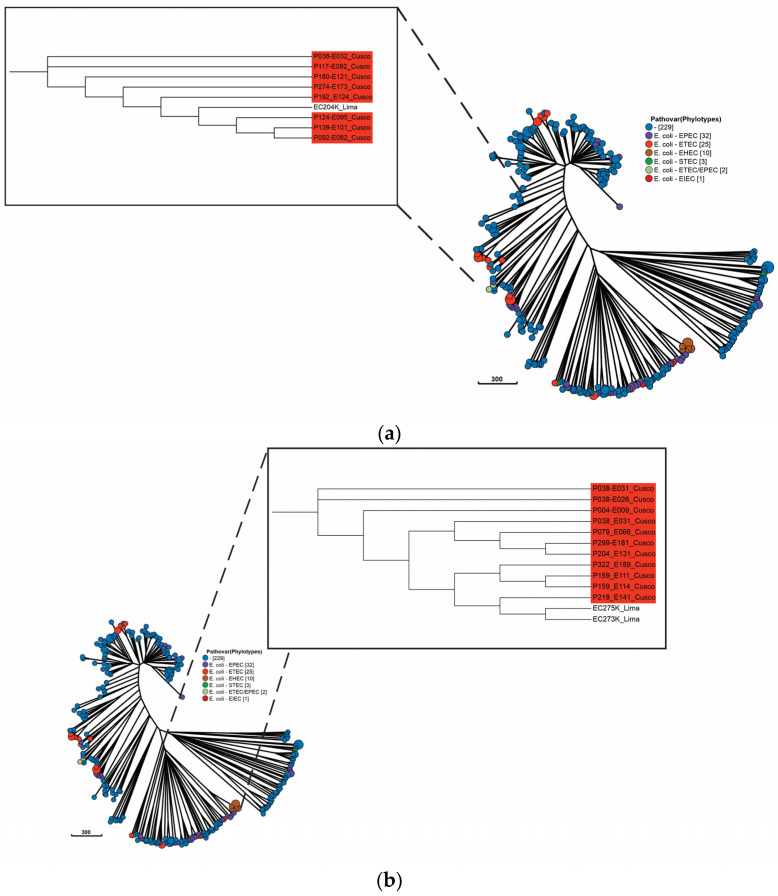
Phylogenetic analysis of genomic sequences obtained from closely related porcine and human *E. coli* isolates. We calculate phylogenies from SNP matrices using a maximum likelihood approach. (**a**) Tree of human ETEC isolates with swine ETEC isolates; (**b**) tree of human EPEC and EHEC isolates with swine EPEC isolate. The human isolates are in red, and the swine isolates are unlabeled. The numbers in brackets are the number of isolates belonging to a pathovar.

**Table 1 antibiotics-12-01748-t001:** Quantity of each gene per swine *E. coli* isolate.

Classes of Antibiotics	Gene	Gene No./Isolate No.
Tetracyclines	*tet(A)*	8/10
	*tet(B)*	4/10
	*tet(M)*	2/10
	*tet(C)*	1/10
Beta-lactam	*bla* _TEM-1A_	2/10
	*bla* _TEM-1B_	4/10
	*bla* _TEM-176_	1/10
	*bla* _CTX-M-14_	1/10
Amphenicol	*floR*	8/10
	*cmlA1*	3/10
Aminoglycosides	*aac(3)-IId*	1/10
	*aadA2*	1/10
	*ant(3″)-Ia*	7/10
	*aph(3′)-Ia_3*	2/10
	*aph(3′)-Ia_1*	2/10
	*aph(3′)-Ia_9*	1/10
	*aph(6)-Id*	3/10
	*aph(3″)-Ib*	3/10
Sulfonamides	*sul3*	8/10
	*sul2*	2/10
Quinolones	*qnrS1*	1/10
	*qnrB19*	9/10
Lincomycin	*lnu(F)*	1/10
	*lnu(G)*	2/10
Trimethoprim	*dfrA12*	2/10
	*dfrA1*	3/10
	*drfA14*	1/10
	*dfrA8*	1/10
Macrolides	*mef(B)*	1/10
	*mdf(A)*	10/10
Colistin	*mcr-1.1*	1/10

## Data Availability

The data used in this study are openly available in at NCBI Genbank, bioproject ID: PRJNA839166.

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
