# Peer review of "Genomic Profiling of Multidrug-Resistant Swine Escherichia coli and Clonal Relationship to Human Isolates in Peru"

_antibiotics, 2023, doi:10.3390/antibiotics12121748_

Round 1

Reviewer 1 Report

Comments and Suggestions for Authors

Genomic Profiling of Multidrug-Resistant Swine Escherichia 2 coli and Clonal Relationship to Human Isolates in Peru

A brief summary

Antibiotics is an international, peer-reviewed, open access journal on all aspects of antibiotics, published monthly online by MDPI with an impact factor of 4.8. The study analyzed enomes and determine their sequence types, serotypes, virulence, and AMR  genes (ARGs) of ten multidrug-resistant (MDR) E. coli isolates from pigs. The relationship was evaluated adding all the available genomes of Peruvian E. coli from humans using the cgMLST+HierCC scheme. Two aEPEC O186:H11-ST29 were identified, of which H11 and ST29 are reported in aEPEC isolates from different sources. An isolate ETEC-O149:H10-ST100 was identified, considered a high-risk clone that is frequently reported in  different countries as a cause of diarrhea in piglets. One ExPEC O101:H11-ST167 was identified, which ST167 is an international high-risk clone related to urinary infections in humans. The authors identified extended-spectrum β-lactamase genes, and one ETEC harboring the mcr-1 gene. CgMLST+HierCC analysis differentiated 3 clusters, in two, the human isolates were grouped with those of swine in the same cluster.

Suggestions

Line 62 - One E. coli isolate have - One E. coli isolate had

Line 93 – correct the letters (a and b) that identify the figures

Line 128 – from farm animals

Line 129 – animals not farms

Lines 131 and 132 – verify and correct the use capital letters (upper-case letters)

Lines 136 and 137 – verify the sentence – suggestion - O149:H10-ST100 strain has been proposed as a clonal lineage considered a high-risk

Line 161- mcr-1 italic

Line 169 – Use resistance instead of showed

Line 169 – 170 – this resistance to antimicrobial classes is phenotypic or genetically?

Line 172 - where of MDR E…

Line 173 – what is RAM?

Line 190 – exclude us

Line 195- population in animals - population of animals?

Line 196- Australia, where analyzed - Australia that analyzed?

Line 219- In summary, isolates from phylogroups other than A and B1 are less common in swine E. coli isolates – do you think that this statement is possible based on ten isolates and isoltes from humans?

Questions

The mcr-1 positive strains, when was it isolated? Highlight the year of isolations and if it is consistent with the emergency in the world and the important of mcr1 positive strains and if colistin is permitted to use in food animals in Peru.

In the methodology section, add the information regarding where the 10 isolates were obtained in the animals (feces, tissues, and if the animals were sick or not).

In the conclusion part, add some impact of the study for the swine production, in terms of diseases, vaccines, if any clone isolated is related to a more severe disease, how the study impact the swine industry and the relation with humans, the public health impact.

Add in the methodology how the MDR was identified in the 10 isolates.

Author Response

Reviewer 1

Line 62 - One E. coli isolate have - One E. coli isolate had was corrected

Line 93 – correct the letters (a and b) that identify the figures was corrected

Line 128 – from farm animals was corrected

Line 129 – animals not farms was corrected

Lines 131 and 132 – verify and correct the use capital letters (upper-case letters) was corrected

Lines 136 and 137 – verify the sentence – suggestion - O149:H10-ST100 strain has been proposed as a clonal lineage considered a high-risk was corrected

Line 161- mcr-1 italic was corrected

Line 169 – Use resistance instead of showed was corrected

Line 169 – 170 – this resistance to antimicrobial classes is phenotypic or genetically?àFurthermore, all isolates carried at least a gene for four different classes of antibiotics (tetracyclines, florfenicol, sulfonamides, and macrolides).It was genetic

Line 172 - where of MDR E…    rewrited

Line 173 – what is RAM?    (AMR corrected)

Line 190 – exclude us corrected

Line 195- population in animals - population of animals? corrected

Line 196- Australia, where analyzed - Australia that analyzed? corrected

  • “Australia” has changed by a study reference (Reid et al.,2019)

Line 219- In summary, isolates from phylogroups other than A and B1 are less common in swine E. coli isolates – do you think that this statement is possible based on ten isolates and isoltes from humans?

  • It was changed by “Therefore, the swine coli isolates analyzed in this study were not grouped into phylogroups other than A and B1”

Questions

The mcr-1 positive strains, when was it isolated? Highlight the year of isolations and if it is consistent with the emergency in the world and the important of mcr1 positive strains and if colistin is permitted to use in food animals in Peru.

  • Methodology was added (2020).
  • A paragraph was added about colistin

In the methodology section, add the information regarding where the 10 isolates were obtained in the animals (feces, tissues, and if the animals were sick or not).

It was described

Add in the methodology how the MDR was identified in the 10 isolates.

It was described

  • Methodology was added

In the conclusion part, add some impact of the study for the swine production, in terms of diseases, vaccines, if any clone isolated is related to a more severe disease, how the study impact the swine industry and the relation with humans, the public health impact.

  • The conclusión was improved

Reviewer 2 Report

Comments and Suggestions for Authors

The provided manuscript "Genomic Profiling of Multidrug-Resistant Swine Escherichia coli and Clonal Relationship to Human Isolates in Peru" obtained information for ten multidrug-resistant (MDR) E. coli isolates from pigs. The Authors described their genomes and determined their sequence types, serotypes, virulence, and antimicrobial genes (ARGs). They proved the presence of high health risk MDR E. coli strains, such as EPEC, ETEC, etc. The study is very interesting, but I have a few remarks and questions.

1) It was not clear to me whether you isolated the tested strains from the pig farm? Did you prove the indicated genes with RT-PCR? I didn't see the gene sequences described anywhere. I recommend describing them in a Table in the section Materials and Methods.

2) Line 72: " 9.6 genes per isolate". I don't understand how one isolate would have 9.6 genes - either 10 or 9! Have you done agar diffusion method or minimum inhibitory concentration of any antibiotic?

3) Figure 2 and 3. The text on the figures is not clear! 

4) The methods are described too briefly and unclearly. I don't see anywhere which reagents you used, how the tested strains were isolated, how the rDNA was isolated, etc. Describe them in more detail and understandably!

Congratulations on a successfully completed research work!

Author Response

Reviewer 2

  • It was not clear to me whether you isolated the tested strains from the pig farm? Did you prove the indicated genes with RT-PCR? I didn't see the gene sequences described anywhere. I recommend describing them in a Table in the section Materials and Methods.

à Methodology was added. (included resistant profile – see suplementary)

  • Line 72: " 9.6 genes per isolate". I don't understand how one isolate would have 9.6 genes - either 10 or 9! Have you done agar diffusion method or minimum inhibitory concentration of any antibiotic?

à it was deleted: “with a mean of 9.6 genes per isolate”

à La evaluación de la sensibilidad antibiótica se hizo mediante la técnica de disco-difusión, se incluyó en la metodología. Los resultados de sensibilidad antibiótica están en la tabla suplementaria S3.

3) Figure 2 and 3. The text on the figures is not clear! 

à the images resolution was improved

  • The methods are described too briefly and unclearly. I don't see anywhere which reagents you used, how the tested strains were isolated, how the rDNA was isolated, etc. Describe them in more detail and understandably!

Methodology this was changed: “Isolation, sequencing and genome sequences”.

Reviewer 3 Report

Comments and Suggestions for Authors

Major comments

1. The writing in the article is not standard and there are many grammatical errors.

2. The writing of the manuscript needs to be improved.

3. The MICs of all isolates must be assayed and showed in manuscript.

4. The gene environments of important genes such as mcr-1 should be characterized.

5. The transferability of important genes such as mcr-1 should be performed.

6. How are these strains isolated.

Minor comments

1. Line 58, ST100 revised as ST-100

2. Line 56, are revised as were

3. Line 52, The abbreviation does not correspond to the interpretation.

4. In table, in Tet(B) , Sul2 and BlaTEM-1A, the first letter of the word should be lowercase.

5. The pixels in figure 3 are too low.

Comments on the Quality of English Language

The writing of the manuscript needs to be improved.

Author Response

Reviewer 3

Major comments

  1. The writing in the article is not standard and there are many grammatical errors.

The manuscript was corrected by an english native writer

  1. The writing of the manuscript needs to be improved.
  2. The MICs of all isolates must be assayed and showed in manuscript.

We did not perform MICs, we performed antibiotic sensitivity test

  1. 4. The gene environments of important genes such as mcr-1 should be characterized.

It was added in methodology, results and discussion.

  1. The transferability of important genes such as mcr-1 should be performed.

The aim of the study was not focus on mcr-1 or some specific antimicrobial resistance, we show a general resistance profile obtained from sequencing.

  1. How are these strains isolated.

Methodology was added

Minor comments

  1. Line 58, ST100 revised as “ST-100” It was corrected
  2. Line 56, “are” revised as “were” It was corrected
  3. Line 52, The abbreviation does not correspond to the interpretation. It was corrected
  4. In table, in Tet(B) , Sul2 and BlaTEM-1A, the first letter of the word should be lowercase. It was corrected
  5. The pixels in figure 3 are too low. ✓ It was improved (please see supplementary)

Round 2

Reviewer 2 Report

Comments and Suggestions for Authors

Dear Authors,

I see the Supplementary materials, but I didn't found the sequences of the tested genes! You have left the text with a mean of 9.6 genes per isolate”.  You clearly don't understand me, there's no way there can be 9.6 genes per isolate! It's like writing that you isolated an average of 9.6 bacterial colonies! Correct it! I can't find the disk diffusion method results attached in the Supplementary materials!

Author Response

Dear reviewer

-In the previous version of the manuscript after the reviewers' comments, We deleted "9.6 genes per isolate" as you recomended. 

-In the Supplementary materials the disk diffusion halos results was added in detail.

-We did not test genes by PCR (virulence, phylogroups and antimicrobial resistance), the results were obtained by whole genome sequencing (WGS) using bioinformatic tools detailed in the methodology. Complete sequences were accepted and are registered in GenBank.